# Impact of fat mass and distribution on lipid turnover in human adipose tissue

Kirsty L. Spalding[1,2], Samuel Bernard[3], Erik Näslund[4], Mehran Salehpour[5], Göran Possnert[5], Lena Appelsved[1], Keng-Yeh Fu[1], Kanar Alkass[1], Henrik Druid[6,7], Anders Thorell[4,8], Mikael Rydén[9] & Peter Arner[9]

Differences in white adipose tissue (WAT) lipid turnover between the visceral (vWAT) and subcutaneous (sWAT) depots may cause metabolic complications in obesity. Here we compare triglyceride age and, thereby, triglyceride turnover in vWAT and sWAT biopsies from 346 individuals and find that subcutaneous triglyceride age and storage capacity are increased in overweight or obese individuals. Visceral triglyceride age is only increased in excessively obese individuals and associated with a lower lipid removal capacity. Thus, although triglyceride storage capacity in sWAT is higher than in vWAT, the former plateaus at substantially lower levels of excess WAT mass than vWAT. In individuals with central or visceral obesity, lipid turnover is selectively increased in vWAT. Obese individuals classified as 'metabolically unhealthy' (according to ATPIII criteria) who have small subcutaneous adipocytes exhibit reduced triglyceride turnover. We conclude that excess WAT results in depot-specific differences in lipid turnover and increased turnover in vWAT and/or decreased turnover in sWAT may result in metabolic complications of overweight or obesity.

[1] Department of Cell and Molecular Biology, Karolinska Institutet, Stockholm SE-17177, Sweden. [2] Metabolism Unit and KI/AZ Integrated Cardio Metabolic Center, Department of Medicine, Karolinska University Hospital, Karolinska Institutet, Stockholm SE 17176, Sweden. [3] Institut Camille Jordan, University of Lyon, Villeurbanne F 69622, France. [4] Department of Clinical Sciences, Danderyd Hospital, Karolinska Institutet, Stockholm SE 18288, Sweden. [5] Department of Physics and Astronomy, Ion Physics, Uppsala University, Uppsala SE-75120, Sweden. [6] Department of Oncology-Pathology, Karolinska Institutet, Stockholm SE 17177, Sweden. [7] Department of Forensic Medicine, The National Board of Forensic Medicine, Stockholm SE 11120, Sweden. [8] Department of Surgery, Ersta Hospital, Karolinska Institutet, Stockholm SE 11691, Sweden. [9] Department of Medicine, Karolinska University Hospital, Karolinska Institutet, Stockholm SE-141 86, Sweden. Correspondence and requests for materials should be addressed to K.L.S. (email: kirsty.spalding@ki.se) or to P.A. (email: peter.arner@ki.se).

Triglyceride turnover in white adipose tissue (WAT) is determined by the balance between lipid storage and lipid removal[1]. Different turnover rates between visceral (vWAT) and abdominal subcutaneous WAT (sWAT) may cause metabolic complications in the overweight/obese state[2–5]. Lipids are mainly stored in white adipocytes by esterification of energy-rich fatty acids and glycerol to form triglycerides. Fatty acids are released from fat cells into the blood stream through triglyceride hydrolysis (lipolysis). Visceral fat cells *in vitro* have increased lipid synthesis and lipolysis compared to abdominal subcutaneous adipocytes, findings that have been corroborated *in vivo* in short-term experimental settings[6–8]. As vWAT is drained by the portal vein, increased levels of portal fatty acids may cause liver-associated metabolic complications, such as hepatic steatosis and/or hepatic insulin resistance, in the overweight and obese states[9–12].

sWAT constitutes most of the body's adipose mass. According to the adipose tissue expandability hypothesis, an attenuated ability to adequately store triglycerides in sWAT could also be of pathophysiological relevance, particularly in abdominal (upper body) obesity[2]. This is at least in part due to an inability to adequately expand the number of fat cells and/or size of the lipid droplet (that is, the subcutaneous fat cell size). In turn, this results in ectopic fat deposition via increased lipid deposition in vWAT and non-adipose tissues such as the liver and skeletal muscle, leading to lipotoxic effects[2,4,5,8] and the development of metabolic complications[13–17]. Thus abdominal sWAT may act as a protective metabolic sink, which enables increased lipid clearance following excess energy intake. Prototypical examples of this pathophysiological process are different forms of lipodystrophies, which are associated with extreme insulin resistance, ectopic lipid deposition and increased risk for cardiovascular disease[4,14–17].

We recently developed a unique method, radiocarbon dating, to investigate triglyceride storage and removal in human WAT under normal living conditions. Above-ground nuclear bomb testing during the cold war period caused a marked increase in atmospheric $^{14}C$ levels (relative to that of the stable $^{12}C$ isotope), which decreased exponentially following the test ban treaty in 1963. By assessing the incorporation of atmospheric $^{14}C$ into fat cell lipids, it is possible to retrospective determine the turnover of lipids during the lifetime of an individual[1]. High or low lipid age reflects low or high triglyceride turnover in WAT, respectively. The rate of lipid removal ($K_{out}$) at a constant fat mass is termed the turnover rate and is expressed as 1/lipid age in years and reflects primarily triglyceride lipolysis[1]. If the size of a lipid depot is known, it is also possible to estimate net lipid uptake ($K_{in}$), which reflects storage capacity of fatty acids into triglycerides and is expressed as kg year$^{-1}$ (ref. 1).

## Results

**Estimating blood and adipose triglyceride age.** Measured lipid $^{14}C$ values for samples collected after 2010 are summarized in Fig. 1. $\Delta^{14}C$ values were not influenced by the individual's year of birth, suggesting that there is no long-lived pool of lipids in fat cells (Fig. 1a). Furthermore, essentially all $\Delta^{14}C$ values were positioned above the atmospheric $\Delta^{14}C$ curve, suggesting that the lipids were older than the collection date and that a valid estimate of lipid age could be obtained (Fig. 1b). This is in contrast to whole or dried blood, which was significantly closer to the sample collection date than adipose lipids from the same subject (Fig. 1b,c and Supplementary Table 1).

**Associations between triglyceride age and anthropometry.** The overall results with triglyceride age were analysed by linear regression (Table 1). They were independent of the subject's age or waist circumference in either adipose region. There was a weak positive linear correlation between triglyceride age and body mass index (BMI) in both depots. Waist-to-hip ratio (WHR, an indirect measure of body fat distribution) correlated negatively with lipid age and only in vWAT. However, even the significant associations were weak (low $r$ values), indicating that there is no simple linear relationship between triglyceride age and total or regional fat mass. In subsequent analyses, we therefore subdivided the subjects into different classes of BMI, body fat mass or body fat distribution.

**Influence of BMI and body fat mass on triglyceride turnover.** When individuals were subdivided into three BMI classes, mean triglyceride age in sWAT was ∼0.6 years higher in overweight and obese subjects (BMI 25–39.9 kg m$^{-2}$) when compared with lean individuals (BMI < 25 kg m$^{-2}$). However, there was no subsequent increase in lipid age in excessively obese subjects (BMI ≥ 40 kg m$^{-2}$) (Fig. 2a). In vWAT, lipid age was not influenced by BMI class, except in excessive obesity where it was ∼0.7 years higher (Fig. 2b). There was no interaction between gender and lipid age in either region (F < 0.4; P > 0.7 by analysis of covariance, graphs not shown). $K_{in}$ was determined in a subgroup of individuals where the size assessments of the investigated WAT depots were available (see Methods section). There was no difference in lipid storage rate ($K_{in}$) between overweight/obese individuals and excessively obese individuals in either adipose depot (Fig. 2c,d). Similar body weight-dependent differences in lipid age and $K_{in}$ were obtained when subdividing the subjects into four, instead of three, separate BMI subclasses (lean, overweight, obese and excessively obese, Supplementary Fig. 1) or quartiles of percentage body fat (Supplementary Fig. 2). As discussed above and shown before[1], $K_{out}$ in subcutaneous fat cells primarily reflects catecholamine-stimulated lipolysis. That this is also true in visceral adipocytes is confirmed by the significant negative correlation between triglyceride age and noradrenaline-induced lipolysis in omental fat cells of 80 subjects (BMI 33.7–57.0 kg m$^{-2}$) (Fig. 2e). There was no correlation between spontaneous (basal) lipolysis and vWAT age. Thus vWAT appears to maintain lipid turnover over a broad range of body fat levels and is only reduced in excessive obesity, primarily due to a diminished triglyceride removal capacity. In sWAT, lipid turnover is attenuated in a similar manner although already in the overweight state.

**Influence of body fat distribution on triglyceride turnover.** The influence of body fat distribution (determined as ratios of android/gynoid fat or visceral/total WAT mass) on triglyceride turnover was determined by subdividing individuals into tertiles and comparing the upper and lower extremes (Fig. 3). While android/gynoid ratio or level of visceral obesity did not influence either subcutaneous triglyceride age (Fig. 3a,e) or $K_{in}$ (Fig. 3b,f), they had a pronounced influence on triglyceride turnover in vWAT (Fig. 3c,d,g,h). Subjects with central obesity (Fig. 3c,d) or pronounced visceral obesity (Fig. 3g,h) had ∼0.7 years younger triglycerides and ∼1.0 kg year$^{-1}$ higher $K_{in}$ than those with peripheral obesity or little visceral fat. WHR (Supplementary Fig. 3) or android/total fat ratios were associated with similar differences in regional triglyceride turnover. BMI adjustment did not influence the findings related to body fat distribution. Thus visceral/central obesity (expressed in any of the four different ways) is linked to increased triglyceride turnover rate (that is, increased capacity to store and mobilize triglycerides) in vWAT only.

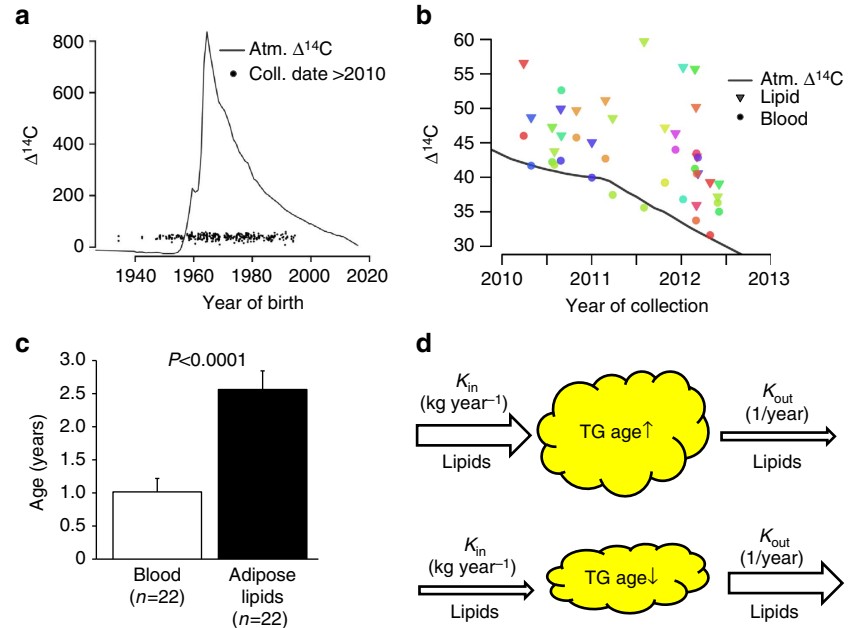

**Figure 1 | Principles of the radiocarbon method. (a)** Lipid $\Delta^{14}C$ values as a function of date of birth for samples collected after 2010. Above-ground nuclear bomb testing during the period of the cold war caused increased atmospheric $^{14}C$ levels relative to stable isotope levels. These values decreased exponentially following implementation of a test ban treaty in 1963. Units are expressed in isotope ratio $^{14}C/^{12}C$ as $\Delta^{14}C$ corrected for decay. Lipid age is determined by averaging lipid $^{14}C$ incorporation into adipose tissue against the atmospheric $^{14}C$-curve (solid black line) over each individual's lifetime to match sample $^{14}C$ levels at the biopsy collection date (grey scatter). Low $\Delta^{14}C$ values indicate that lipids have a high turnover rate ($K_{out}$), or equivalently that lipid residency times (lipid age) are short, with the relationship age $= 1/K_{out}$. **(b)** Donor-paired sWAT lipid (triangles) and blood (filled circles) versus $\Delta^{14}C$ levels as function of sample collection date. Each colour corresponds to a single donor. Lower blood $\Delta^{14}C$ levels indicate a higher turnover rate. **(c)** Summary data showing the average age of blood and lipid. Values are mean ± s.e.m. and compared by paired $t$-test. **(d)** Lipid age is used to assess triglyceride turnover as detailed in the Methods section. The main factors regulating triglyceride turnover in fat cells are the storage capacity ($K_{in}$) and removal capacity ($K_{out}$) of fatty acids. Such processes include fatty acid uptake, esterification, mobilization/re-esterification and oxidation. In an expanding adipose tissue, $K_{in} > K_{out}$, whereas the reverse is observed when adipose tissue mass is lost.

**Table 1 | Correlation between age, body mass index, waist circumferences or waist-to-hip ratio for visceral or subcutaneous lipid age.**

| Regressor | Visceral | | Subcutaneous | |
|---|---|---|---|---|
| | $r$-value | $P$ value | $r$-value | $P$ value |
| Age | 0.05 ($n=253$) | 0.47 | 0.05 ($n=329$) | 0.36 |
| Body mass index | 0.17 ($n=255$) | 0.007 | 0.13 ($n=331$) | 0.019 |
| Waist circumference | 0.03 ($n=206$) | 0.65 | 0.01 ($n=203$) | 0.91 |
| Waist-to-hip-ratio | $-0.21$ ($n=201$) | 0.003 | 0.03 ($n=198$) | 0.64 |

$n$, number of subjects. Simple linear regression analysis was used.

To further elucidate the relationship between lipid age and regional body fat distribution, we selected the obese subjects where detailed information on sWAT and vWAT amounts as well as other fat distribution parameters were available (see Methods section). They were subdivided according to quartiles of lipid age in the two depots and we compared the upper and lower quartiles regarding body fat mass and distribution (Table 2). The subdivision according to sWAT lipid age showed no significant differences between the groups. In contrast, low vWAT triglyceride age was associated with increased WHR and central/visceral fat accumulation despite a somewhat lower BMI/total fat mass.

To investigate regional differences in triglyceride turnover between sWAT and vWAT, we compared triglyceride age in

paired samples (Fig. 4). The average subject BMI was $38 \pm 7 \, \mathrm{kg \, m^{-2}}$ in Fig. 4a and $40 \pm 5 \, \mathrm{kg \, m^{-2}}$ in Fig. 4b. Triglyceride age was slightly higher ($\sim 0.3$ years) and triglyceride storage much higher ($\sim 0.7 \, \mathrm{kg \, year^{-1}}$) in subcutaneous vs visceral adipose tissue.

**Influence of fat cell size on sWAT triglyceride age.** We tested the sWAT expandability hypothesis[2] by comparing sWAT fat cell size with sWAT triglyceride age in a subgroup of individuals with 'healthy' or 'unhealthy' obesity where we had available clinical data to allow scoring according to the ATPIII criteria (see Methods section). As expected, despite similar BMI and abdominal sWAT mass between the groups, unhealthy obese subjects had fewer but larger fat cells compared with healthy individuals (Supplementary Table 2). Results with lipid age are shown in Fig. 5. There was no interaction between fat cell size and triglyceride age in any group. However, when subjects were grouped into small (476–902 pl) or large (904–1,420 pl) fat cells, triglyceride age was increased in the small fat cells of unhealthy compared to healthy subjects. No such differences were observed in subjects with large fat cells. These results could not be explained by differences between unhealthy and healthy obese on the basis of fat cell size.

**Discussion**
This study addresses an issue that has not been possible to answer previously, namely, whether *in vivo* lipid handling in visceral and abdominal sWAT differs in humans. We confirm that subcutaneous triglyceride age and triglyceride storage capacity are increased in overweight or obese subjects[1,18] and additionally

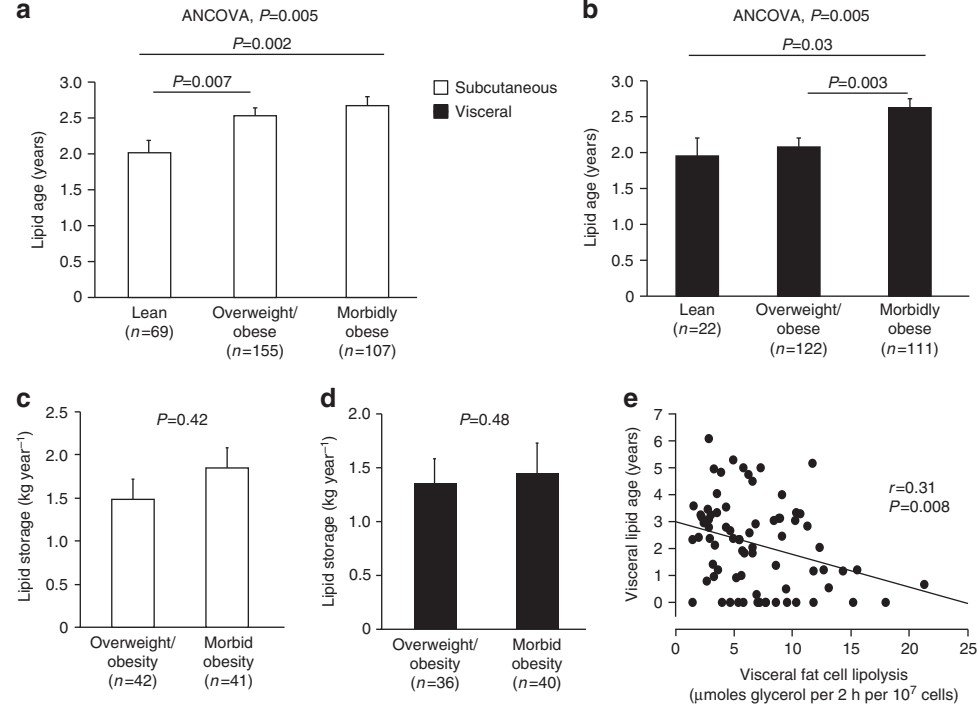

**Figure 2 | $^{14}$C lipid age and lipid storage in different BMI groups. a,c** Are subcutaneous and **b,d** are visceral adipose tissue. Values are mean ± s.e.m. and are compared using analysis of variance and a *post hoc* test. $n$ = number of subjects. **e** Is a comparison between lipolysis and lipid age in the visceral region in 80 individuals using linear isoregression.

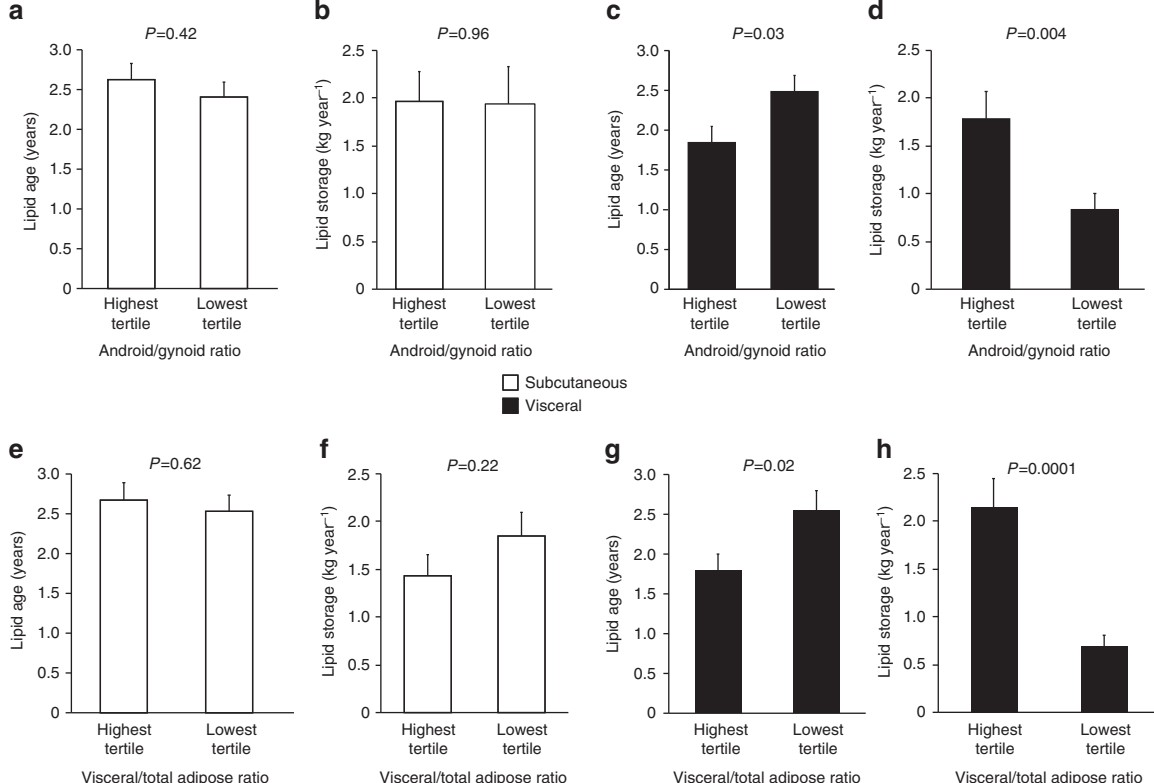

**Figure 3 | $^{14}$C lipid age and lipid storage in different adipose distribution groups.** Individuals ($n$ = 148) who underwent DEXA were divided into tertiles based on (**a–d**) android/gynoid ratio (upper panels) and (**e–h**) ratio of visceral/total fat mass (lower panels). Visceral adipose tissue depicted with black bars and subcutaneous with open bars. Values are mean ± s.e.m. for the highest and lowest tertile and are compared using an unpaired *t*-test.

**Table 2 | Clinical phenotypes of subjects with upper or lower 25th percentile values for adipose lipid age.**

| Phenotype | Subcutaneous ($n = 104$) | | | Visceral ($n = 104$) | | |
|---|---|---|---|---|---|---|
| | Lower | Upper | *P*-value | Lower | Upper | *P*-value |
| Age, years | 41 ± 10 | 42 ± 10 | 0.40 | 41 ± 11 | 40 ± 10 | 0.54 |
| Body mass index, kg m$^{-2}$ | 41 ± 6 | 41 ± 6 | 0.89 | 39 ± 7 | 41 ± 5 | 0.02 |
| Waist-to-hip ratio | 1.01 ± 0.07 | 0.99 ± 0.09 | 0.33 | 1.02 ± 0.08 | 0.97 ± 0.08 | 0.002 |
| Gynoid fat by DEXA, kg | 9.3 ± 2.6 | 8.9 ± 1.8 | 0.32 | 8.4 ± 1.9 | 9.6 ± 2.0 | 0.85 |
| Total body fat by DEXA, kg | 58 ± 11 | 57 ± 12 | 0.69 | 59 ± 9 | 58 ± 11 | 0.06 |
| Android/gynoid fat mass, ratio | 1.15 ± 0.13 | 1.14 ± 0.13 | 0.86 | 1.17 ± 0.15 | 1.10 ± 0.10 | 0.02 |
| Visceral fat mass, kg | 2.2 ± 0.8 | 2.3 ± 1.0 | 0.71 | 2.5 ± 1.3 | 2.0 ± 0.7 | 0.05 |
| Abdominal subcutaneous fat mass in the biopsy region, kg | 3.8 ± 1.1 | 3.6 ± 1.2 | 0.54 | 3.3 ± 0.9 | 3.7 ± 1.0 | 0.06 |
| Visceral adipocyte index (visceral fat mass/total fat mass) | 0.039 ± 0.013 | 0.041 ± 0.015 | 0.66 | 0.046 ± 0.018 | 0.035 ± 0.012 | 0.004 |

DEXA, dual X-ray absorptiometry.
Values are mean ± s.d. and compared by unpaired *t*-test. Lower 25th percentile cutoff value for lipid age was 1.83 years in subcutaneous adipose tissue and 1.45 years in visceral adipose tissue.
Corresponding values for upper 25th percentile were 3.49 and 3.17 years, respectively.

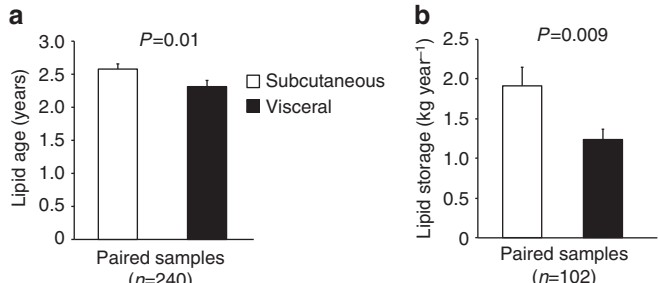

**Figure 4 | Regional differences in lipid turnover.** (**a**) [14]C lipid age and (**b**) lipid storage in subcutaneous (open bars) versus visceral (black bars) adipose tissue. Values are mean ± s.e.m. for paired samples and are compared using a paired *t*-test. *n* = number of pairs.

show that the 0.6 year increase is similar over the entire of spectrum of body fat excess. This is in contrast to the visceral region, in which average triglyceride age remains unaltered at all BMI or percentage body fat levels, except among the extremely obese where it is increased. In the visceral region, as previously demonstrated in the subcutaneous region[1], triglyceride age is determined by lipolysis (high age is linked to a low lipolysis rate and vice versa). Thus abdominal sWAT seems to attain a maximal capacity to store and release lipids already in the transition from the lean to the overweight state. This may explain why lipids accumulate in the liver following only a moderate body weight increase in non-obese subjects[19,20]. In contrast, vWAT appears to maintain lipid turnover over a broad range of body fat levels and is only reduced in morbid obesity. Nevertheless, in both

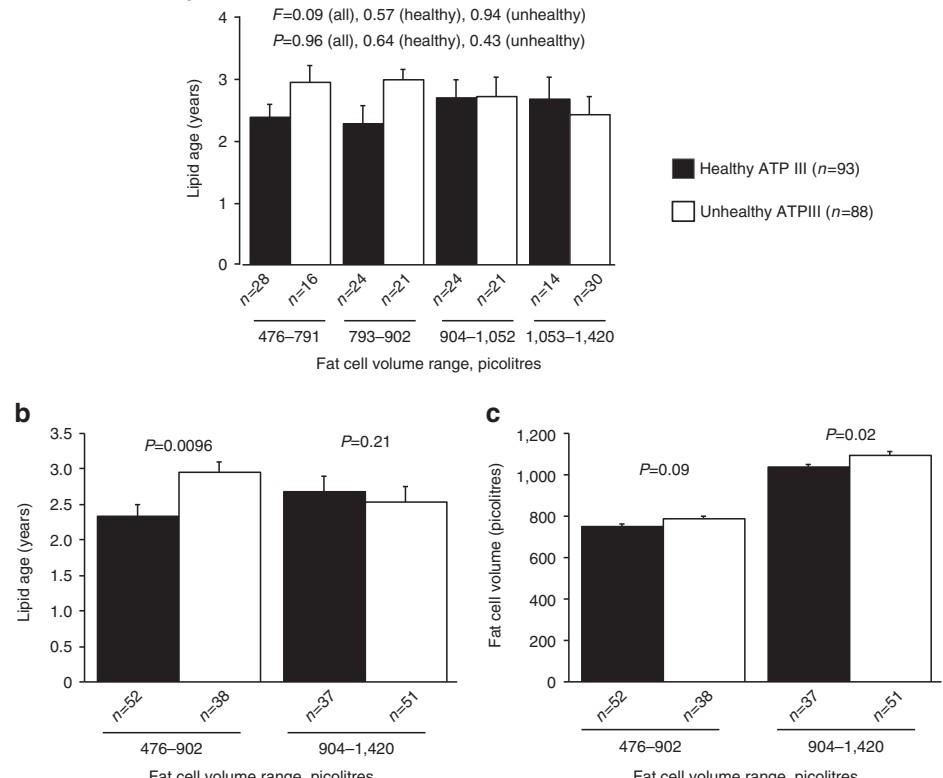

**Figure 5 | Impact of fat cell volume on lipid age in sWAT.** (**a**) Differences in lipid age across different fat cell volumes in obese healthy or unhealthy subjects. (**b**) Lipid age in healthy or unhealthy obese with either small or large fat cells. (**c**) Fat cell volume in healthy or unhealthy obese. Values are mean ± s.e.m. and compared by analysis of variance or unpaired *t*-test. Number of individuals are indicated below each bar.

depots diminished triglyceride removal capacity constitutes the major mechanism explaining the reduction in lipid turnover.

Studies using isotope tracers show that subjects with upper body obesity have a higher capacity to mobilize fatty acids from the splanchnic (that is, visceral fat) region than those with lower body obesity[21]. In contrast, body fat distribution has no impact on fatty acid mobilization from the non-abdominal (that is, femoral) adipose region[21]. As reviewed in detail, the latter WAT region is much less metabolically active than the abdominal sWAT or vWAT regions[6,21]. These data indicate that body fat distribution primarily influences visceral lipid turnover. Our present data show that this is indeed the case. Subjects with visceral obesity (expressed in any of the four ways) have increased triglyceride turnover rate. This is attributed to a combination of increased capacity to store and mobilize triglycerides. The impact of body fat distribution is selective for vWAT, as subcutaneous triglyceride turnover is not influenced by the different measures of adipose distribution. Supported by the findings in Table 2 and Fig. 3, the higher vWAT triglyceride turnover in visceral/upper body obesity can be expected to increase portal fatty acid fluxes, facilitating metabolic deterioration according to the mechanisms discussed above.

We observed that lipid age was significantly higher in subcutaneous than visceral adipose tissue in obese subjects. This cannot be explained by regional differences in storage of very-low-density lipoprotein-derived older fatty acids as fatty acid recycling is similar in both regions in non-obese or obese subjects[22]. Instead, the overall capacity to store lipids is much higher in the subcutaneous than in the visceral region, which could be an important reason why sWAT is the major site for lipid storage in obesity.

Our findings may also be important in supporting the subcutaneous adipose expandability hypothesis[2], although in a somewhat unexpected way. On the basis of adipose morphology, our findings are congruent with the expandability hypothesis as unhealthy obese subjects had fewer but larger fat cells than healthy obese matched for BMI or fat mass. Regarding lipid turnover, one would intuitively expect increased lipid age in unhealthy obese subjects with few but large subcutaneous adipocytes. However, this was not the case as a reduced turnover was only observed in unhealthy subjects having relatively small fat cells. This adds yet another perspective to the sWAT expandability. It suggests that ectopic fat deposition may not only result from an insufficient increase in subcutaneous fat cell size and/or number but also to attenuated triglyceride turnover. This develops before the subcutaneous cells have attained their maximum size. In larger fat cells, other mechanisms may explain the link to ectopic fat.

During the relatively short lifespan ($\sim$10 years) of an adult human adipocyte, its triglycerides are turned over $\sim$6 times[15,23]. Even relatively minor changes in this rate could play a pathophysiological role in the development of metabolic disturbances[1,24,25]. For example, lipid age in sWAT is significantly decreased by $\sim$0.3 year in familial combined hyperlipidaemia[15]. In the present study, we had sufficient statistical power to detect a 0.25 year difference. The 0.3–1.0 year difference in triglyceride age between adipose depots and different categories of body fat mass or distribution observed herein could therefore be clinically relevant.

Our study has some limitations. As we only investigated Caucasians, we cannot rule out the possibility of ethnic variations in regional triglyceride turnover. Recently (a long time after this study started in 1995), it has become apparent that gluteal/femoral WAT may have protective effects against metabolic complications of obesity[26]. As this information was not available when this study was designed and during a long initial period of

recruitment, we did not sample tissue from the gluteal/femoral region. Because of the aforementioned problems in obtaining sufficient amounts of sample, it was not possible to relate [14]C findings to clinical chemistry variables. Some caution should also be taken in interpreting the [14]C data. For example, extreme diets such as almost exclusive marine diets may lower the intake of atmospheric [14]C and thereby the incorporation into triglycerides[27]. However, after the year 2000, when most of the present samples were collected, the levels of atmospheric [14]C have been low implying that the effects of exclusively marine diets are very small[27]. In addition, these diets are very rare (almost non-existent) in urban Swedish populations such as the one presently investigated[28]. [14]C content may also differ between fresh and stored food products, resulting in a so-called dietary lag. We show that lipid age is significantly older than blood taken from the same individual, demonstrating a significant period in which lipid is resident in the adipose tissue and not merely a result of a dietary lag. While worth a mention, these methodological issues are less relevant for the present study, which was focussed on intra-individual, instead of inter-individual, comparisons of sWAT and vWAT samples.

In summary, triglyceride turnover in WAT is subjected to marked regional variations. sWAT has a greater storage capacity than vWAT but the former reaches a maximal level already in the overweight state, which may increase the risk of lipid spillover and ectopic fat deposition. Furthermore, only visceral adipose tissue is influenced by body fat distribution, where both storage and removal of triglycerides are increased in subjects with visceral or upper body obesity. These region-specific differences in lipid turnover may help to explain why upper-body fat distribution and visceral fat accumulation are closely linked to metabolic complications. In addition, inappropriate subcutaneous lipid handling in small adipocytes may be a factor underlying the development of an unhealthy obese phenotype. Admittedly, further studies combining detailed clinical phenotyping with measures of regional adipose lipid turnover are needed to firmly establish the role of regional adipose triglyceride turnover in metabolic disease.

## Methods

**Study participants.** We included 320 patients scheduled for elective cholecystectomy for symptomatic, uncomplicated gall-stone disease (no jaundice) or bariatric surgery for obesity and 26 deceased subjects (see below). A clinical summary is given in Supplementary Table 3. Because we did not know in advance if sufficient amounts of WAT could be obtained, in particular paired samples of vWAT and sWAT from lean or overweight patients, it was not possible to standardize patient selection or perform blood sampling and clinical examinations on a separate occasion preoperatively besides BMI measures. In 210 patients (of which 20 were lean or overweight as defined below), we also had preoperative measures of waist and hip circumferences, which were used to calculate WHR. In 181 obese subjects described in Supplementary Table 2, we had clinical information on fasting blood glucose, blood pressure, serum lipids and serum circumference. These measures were used to calculate ATPIII score as described[29]. Scores 0–2 were considered as healthy and 3–5 as unhealthy. In the same subjects, the sWAT mean fat cell size was determined as described in detail previously[30]. In 148 subjects (only 4 were lean or overweight), we also performed dual X-ray absorptiometry (DEXA) as described[31]. In brief, body fat composition was assessed using a GE Lunar iDXA and the software enCORE (version 14.10.022)[32] provided by the manufacturer (GE Healthcare, Madison, WI, USA). From the DEXA measurement, it is possible to determine total fat, abdominal (android) fat, peripheral (gynoid) fat and estimated visceral fat (EVAT) in the android region[33]. Determination of EVAT with this DEXA method has a strong correlation ($r > 0.95$) with measures using computed tomography[33]. Estimated subcutaneous adipose tissue in the android region (ESAT) was calculated as total android fat minus EVAT. The subcutaneous fat biopsy in the present study was obtained from the abdominal wall at the same level as the measured ESAT. Importantly, EVAT and ESAT do not measure the total visceral or abdominal subcutaneous adipose tissue but a representative segment of each of these regions. Upper body fat distribution was determined as the android/gynoid ratio by DEXA and visceral adiposity as the ratio of visceral/total adipose tissue mass by DEXA. Because access to sufficient amounts of WAT was difficult in lean and overweight subjects, we also obtained WAT samples from

26 recently deceased subjects, out of which 20 were lean/overweight (details in Supplementary Table 4). None of the included subjects had any known malignancies. To compare whole blood turnover with lipid turnover from the same subject, whole blood and subcutaneous abdominal adipose biopsies were taken from 22 subjects; from 4 of these, dried blood was also obtained. Further information can be found in Supplementary Table 1.

WAT was obtained from the greater omentum (visceral) and the abdominal subcutaneous region (Supplementary Table 3). In total, WAT from 345 subjects was obtained for $^{14}$C-analysis and paired visceral/subcutaneous adipose samples were available from 241 of these individuals. Body fat status was classified in two ways. We used World Health Organization criteria: lean (BMI = 18–24.9 kg m$^{-2}$), overweight (BMI = 25–29.9 kg m$^{-2}$), obese (BMI = 30–39.9 kg m$^{-2}$), and morbidly obese (BMI ≥ 40 kg m$^{-2}$). We also used quartiles of percentage body fat, which was estimated by a formula based on age, sex and BMI[34]. We have previously compared formula-derived percentage body fat with direct measures with either DEXA or bioimpedance[35]. Both direct measures correlate strongly with the formula measure ($r > 0.9$, slope almost 1.0 and intercept not significantly different from zero). The study was approved by the regional ethics board at Karolinska Institutet and performed in accordance with the statutes in the Declaration of Helsinki. It was explained in detail to each living participant and informed written consent was obtained. A very small number of subjects received oral treatment for type 2 diabetes or hyperlipidaemia (Supplementary Table 3). Results were not significantly altered if these subjects were excluded from the analyses.

**Extraction of lipids from adipose tissue preparations.** Samples were collected between 1995 and 2014 and stored at −70 °C. They were homogenized and lipids were extracted exactly as described[1]. The resulting lipid sample was processed for $^{14}$C measurements as described before[17]. As discussed previously[1], the extracted lipids mainly represent triglycerides in fat cells. Because the mean triglyceride age in human fat cells is about 2 years and the half-life of $^{14}$C is ~5,700 years[1], preoperative treatment, surgical or anaesthetic procedures, sample storage or death do not affect triglyceride age measurements.

**Carbon isotope ratio measurements.** Whole blood (either processed as defrosted frozen blood or dried blood) and lipid extracts were subjected to carbon isotope ratio measurements using accelerator mass spectrometry as described previously[1]. In brief, a dedicated sample preparation method was developed, which facilitated analyses of samples with total carbon masses from mg to μg[36]. An important feature of the method is the low amount of stray carbon introduced into the samples, reducing sample-to-sample fluctuations and improving measurement accuracy. All samples were prepared by initial conversion into $CO_2$ gas and subsequent reduction into graphite, as described below. CuO was used as the oxidizing agent, which was added to the samples in quartz tubes. The tubes were subsequently evacuated and sealed with a high temperature torch. The quartz tubes were placed in a furnace set at 900 °C for 3.5 h to combust all carbon to $CO_2$. The formed gas was cryogenically purified and trapped. The collected $CO_2$ gas was then reduced into graphite in individual sub-ml reactors at 550 °C for 6 h in the presence of zinc powder as reducing agent and iron powder as catalyst. The graphite targets, all containing approximately 500 μg of carbon, were pressed into target holders and measured at the Department of Physics and Astronomy, Ion Physics, Uppsala University, using a 5 MV Pelletron tandem accelerator[37]. Stringent and thorough laboratory practice was implemented to minimize the introduction of contaminant carbon into the samples such as: preheating all glassware and chemicals under oxygen flow prior to samples' preparation to eliminate surface-absorbed $CO_2$. In addition, the $CO_2$ from the samples were split and a small fraction (50 μg C) was used to measure the isotopic ratio δ$^{13}$C by stable isotope ratio mass spectrometry, presented as per mille deviation from a reference sample. All $^{14}$C data are $^{14}$C/$^{12}$C ratios reported as decay-corrected Δ$^{14}$C in per mille deviation from a standard[38] or Fraction Modern[39]. The measurement error was determined for each sample and is typically in the range of ±8 to 12‰ (2 s.d.) Δ$^{14}$C. All AMS analyses were performed blind to age and origin of the sample. Blood and fat cell triglyceride age *in vivo* was estimated using the measures of $^{14}$C in blood and lipid set in relation to atmospheric levels of $^{14}$C at the time of sample collection. For lipid measurements, the method predominantly (95%) measures the age of the fatty acid moiety of triglycerides. Lipid age and DEXA measures of visceral/abdominal subcutaneous fat mass allowed us to quantify $K_{in}$ for the two adipose regions (kg year$^{-1}$).

**Estimating lipid age.** The method used here is based on a $^{14}$C dating method for biological samples[1,23,40,41]. Atmospheric $^{14}$C is continuously integrated into the food and the lipids we consume. Because most of the lipid mass is stored in adipocytes, the $^{14}$C distribution in adipocytes should reflect the residence time of the lipids in the body. In order to relate changes in $^{14}$C content over time with lipid aging and renewal, we used a linear partial-differential equation structured in age:

$$\frac{\partial f(t,a)}{\partial t} + \frac{\partial f(t,a)}{\partial a} = -K_{out} f(t,a),\tag{1}$$

$$f(t,0) = K_{in},\tag{2}$$

$$f(0,a) = 0.\tag{3}$$

The parameter $K_{in}$ is the net lipid uptake by the adipose tissue (units: kg year$^{-1}$). The parameter $K_{out}$ controls the rate at which lipids are replaced (units: 1 year$^{-1}$). In equation (1), the left-hand-side terms are conservation terms stating that lipids advance in age at the same rate as time. The right-hand-side term accounts for the removal of lipids. The boundary condition describes the constant uptake of new lipids $K_{out}$ (equation 2) To specify the model completely, the initial age distribution of lipids at subject age $t = 0$ was set to 0 (equation 3), meaning that the initial fat mass was negligible. When the total fat mass stays approximately constant (not varying more than few kg over a year), the fat mass is considered to be at steady state, and $K_{out}$ is termed turnover rate. The function $f(t, a)$ is the density of lipids of age $a$ at time $t$ ($f$ has units cells year$^{-1}$). The density $f$ related to the total body fat mass $F$ in the following way:

$$F(t) = \int_0^t f(t,a)da.\tag{4}$$

The model defined by equations (1–3) is a one-compartment model in which lipids enter adipose tissue at age 0 and enter only once; when they leaves the adipose tissue, lipids are not recycled.

For each lipid sample $(i, r)$ corresponding to subject $i$ and adipose region $r$ ($r$ is subcutaneous or visceral), two independent measurements are available: the total fat mass ($F_i$) of the donor and the $^{14}$C abundance in the lipid sample $C_i$. We assumed that the following two hypotheses were valid over a few years before the collection date, corresponding to the time needed to turnover most of the lipids.

H1. The turnover rate $K_{out}$ of each lipid sample is constant.

H2. The weight $F$ of each donor is constant.

Based on previous results[1], lipids are generally < 4 years. Even considering that these hypotheses are not always satisfied, the characteristic timescale for most of the donors is a few years.

With H1–H2, it is possible to get a direct estimate of the turnover $K_{out}$ for each lipid sample. The $^{14}$C abundance predicted by the model for a lipid sample collected at time $t_d$ is

$$\tilde{C}(K_{out}) = \frac{\int_0^{t_d - t_b} K(t_d - a)f(t,a)da}{F(t)}\tag{5}$$

The value $t_b$ is the time of birth of the subject. The function $K$ is the atmospheric $^{14}$C abundance curve[3], or the 'atmospheric curve'. Because of hypothesis H2 and the presence of the ratio between $f$ and $F$ in equation (5), the parameter $K_{out}$ does not appear explicitly and the only parameter remaining is $K_{out}$. Together with H1, we obtain that the prediction $\tilde{C}$ depends only on the constant turnover rate $K_{out}$. The turnover $K_{out}$ was calculated based on equation (5) with Matlab (Naticks, MA) using the routine fzero.

For each $^{14}$C sample $C^{(i, r)}$ corresponding to subject $i$ and apdipose region $r$, we solved the equation

$$\tilde{C}\left(K_{out}^{(i,r)}\right) - C_i = 0,\tag{6}$$

for the turnover rate $K_{out}^{(i,r)}$. Equation (6) has usually one or two solutions, because of the unimodal nature of the atmospheric curve. When two solutions were found, the highest turnover rate was selected as the correct solution. Low turnover solutions were considered as having no physiological reality, as they can be found only in subjects born before the rise of the $^{14}$C levels around 1955. When $C^{(i, r)}$ was below contemporary levels, no turnover or only a low turnover corresponding to the rising part of the atmospheric curve could be found. In that case, we assumed that the sample was contemporary and assigned an infinite turnover rate to it.

Based on the turnover rates found, the average age of lipids with turnover rate $K_{out}$ in a donor aged $t$ was calculated as

$$\langle a \rangle_t^{(i,r)} = \frac{-t \exp\left(-K_{out}^{(i,r)}t\right) + \frac{1 - \exp\left(-K_{out}^{(i,r)}t\right)}{K_{out}^{(i,r)}}}{1 - \exp\left(-K_{out}^{(i,r)}t\right)}.$$

For samples with infinite turnover rates, $\langle a \rangle_t = 0$ year.

**Estimating lipid uptake.** Total body fat mass $F(t)$ evolves according to a balance equation, obtained by integrating the partial-differential equation (equation (1)),

$$\frac{dF(t)}{dt} = K_{in} - K_{out} F(t),\tag{7}$$

where $K_{in}$ is the net lipid uptake by the adipose tissue (units: kg year$^{-1}$) and $K_{out} F(t)$ is the net lipid removal rate (units: kg lipids year$^{-1}$), as defined above.

When H1–H2 are assumed, the left-hand side of equation (7) is zero. This implies that the body fat mass is determined by the balance between lipid storage and lipid turnover:

$$F = \frac{K_{in}}{K_{out}}.\tag{8}$$

An alternate expression of the total body fat mass with respect to lipid uptake and average lipid age is

$$F = K_{in} \langle a \rangle_t.\tag{9}$$

The lipid uptake $K_{in}^{(i,r)}$ in subject $i$ and in adipose region $r$ with adipose region fat mass $F^{(i,r)}$ is then estimated as

$$K_{in}^{(i,r)} = \frac{F^{(i,r)}}{\langle a \rangle_t^{(i,r)}}. \qquad (10)$$

**Lipolysis measurements.** In 80 subjects, omental fat cell lipolysis was measured exactly as described[42]. In brief, fresh adipose samples were processed to isolate fat cells, which were subsequently incubated *in vitro* for 2 h in the absence or presence of different concentrations of noradrenaline (the major lipolytic hormone in humans). Glycerol release (lipolysis index) per number of incubated fat cells at the maximum effective concentration of noradrenaline is reported.

**Statistics.** Values are mean ± s.e.m. in figures and s.d. was used for dispersion in the text. They were compared by linear regression, paired or unpaired *t*-test and analysis of variance with Fisher's protected least significant difference as *post hoc* test. Analysis of covariance was used to investigate interaction between gender and lipid age results. We used previously published data on triglyceride age in sWAT for an initial statistical power calculation. For regional differences, we assumed an s.d. of triglyceride age of 1.4 years. To detect a 0.25 year difference with 80% statistical power at $P < 0.05$, 241 paired samples were needed.

**Data availability.** The data that support the findings of this study are available from the corresponding authors upon reasonable request.

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

## Acknowledgements

We are grateful for the preparation of samples for accelerator mass spectrometry by Peter Senneryd. This study was supported by grants from European Research Council (ERC), Swedish Research Council, Novo Nordisk Foundation (including Tripartite Immuno-metabolism Consortium (TrIC), Grant Number NNF15CC0018486), CIMED, Swedish Diabetes Foundation, Stockholm Community, Karolinska Institutet, The Erling-Persson Family Foundation, KI-AZ ICMC, the Vallee Foundation and the Diabetes Research Program at Karolinska Institutet. We are grateful for the critical input to the manuscript provided by Professor Keith Frayn, Oxford, UK.

## Author contributions

P.A. and K.L.S. designed the study. L.A., K.-Y.F. and K.A. prepared lipid samples. M.S. and G.P. measured $^{14}C$. S.B. performed analysis of lipid age and turnover. E.N., H.D. and A.T. selected and phenotyped study subjects together with P.A. and M.R. P.A. performed statistical analysis. P.A., K.L.S., S.B. and M.R. wrote together the first version of the manuscript. All authors contributed to the subsequent writing and approved the final version of the manuscript.
