## [Peer Review File · Nature Communications]

Reviewer #1 (Remarks to the Author)

The manuscript by Spalding et al. describes an approach using atmospheric ^{14}C incorporation to measure triglyceride age, and by extension, storage capacity and turnover in human subjects. This method was then applied to a few hundred subjects across a range of body weight and types of fat distribution, with findings correlated with clinical variables. This work builds on the seminal studies done by this group using atmospheric ^{14}C to date adipocytes and other cell types. This work, as applied here, has the potential to provide insight into the mechanisms underlying metabolic dysfunction in human obesity. While this reviewer finds no specific faults with the data shown here, the manuscript as written is almost entirely descriptive. Moreover, in its current form, this manuscript is not likely to be of interest to the broad readership of Nature Communications and would be better suited for a subspecialty journal.

The following are specific points of criticism:

1. The study is very descriptive with little mechanistic material. The authors might have considered pre-clinical studies in mouse models, which would have allowed for direct measures of lipolysis in each animal as well as specific examination of tissue morphology, gene expression, and other phenotypic analyses.
2. The paper as written does not present broad conclusions or novel approaches that are likely to be of broad interest to the readers of this journal. While this would certainly appeal to those with a focus in metabolic research, unless extensively rewritten, the manuscript would not be of clear interest to those from other scientific disciplines.
3. The correlation shown in Figure 2E does not appear especially impressive. There are subjects with low visceral lipid age and low rates of lipolysis, and there are also subjects with high visceral age and high rates of lipolysis. The manuscript may have been more informative if the authors had talked about some of these outliers and possibly characterized the phenotypes on either extreme.

Reviewer #2 (Remarks to the Author)

Spalding and coworkers tested the hypothesis that different lipid (triglyceride) turnover rates between visceral and abdominal subcutaneous (SC) adipose tissue (AT) maybe related to adverse metabolic outcomes of obesity in individuals with overweight or obesity. So far, no data are available to formally proof this hypothesis - mainly due to a lack of appropriate methods. Therefore data presented in the manuscript are novel and add significantly to our understanding of how different fat depots handle lipids distinctly and how that may contribute to obesity comorbidities. The authors took advantage of their experience with measuring atmospheric ^{14}C incorporation into lipids in an unique and innovative setting of experiments. In the context of a large cross-sectional study of 346 individuals with a wide range of BMI and metabolic alterations, triglyceride age has been assessed using the ^{14}C incorporation method. The authors found that SC triglyceride age and storage capacity were increased in overweight/obesity, whereas visceral triglyceride age remained unaffected by fat mass. Formally, the authors found evidence for the SC expandability theory by showing that individuals with central/visceral obesity have a higher triglyceride storage capacity in visceral fat. Data are novel, the study concept is innovative and timely and the manuscript is well written. There are a few comments, which should be addressed:

- 1) In the abstract, a link between visceral triglyceride storage capacity and/or ineffective SC storage with metabolic consequences should be included into the "results" part.

2) The authors show that all ^{14}C values were positioned above the atmospheric curve suggesting that the lipids were older than the collection date. Is there another way to Support that assumption, e.g. by testing circulating blood cell ^{14}C incorporation in parallel to the adipose tissue measurements? Maybe it would be sufficient to demonstrate differences between "short lived" and "older" cells in a small subgroup.

3) The statement that there were no or only weak positive linear correlations between BMI or WHR and lipid age is difficult to follow, give that no data are shown and the p-values indicate at least in part formal significance. The authors should clarify that.

4) It would be interesting to see, whether categorization of individuals by lipid turnover rate unravels (e.g. upper and lower 10%) associations with distinct phenotypes, such as obesity-related metabolic alterations, fat distribution, BMI etc.

5) Metabolically healthy obesity has been indirectly linked to SC AT expandability. Is there any way to use the data to test this hypothesis in the conext of metabolically healthy versus unhealthy obese?

Reviewer #3 (Remarks to the Author)

Title: IMPACT OF FAT MASS AND DISTRIBUTION ON LIPID TURNOVER IN HUMAN ADIPOSE TISSUE

This work uses the ^{14}C bomb-pulse dating method to assess the turnover rate, of the order of a few years, of human fat tissue. While this has been done in previous work by the authors the present paper deals in particular with assessment of the small, but sometimes significant, differences found between different types and locations of fat tissue and dependence on individual lean/obese body constitution.

Being a specialist only in the field of ^{14}C dating methods, I cannot judge if the results and conclusions are of sufficient wider interest to warrant publication in Nature Communications. However, bomb-pulse dating has revolutionised our knowledge of (human) tissue turnover, spanning from fast (e.g. fat tissue) to basically inert tissue, such as eye lenses, dentine, tendon and cartilage collagen, and the present paper's use of the method appears solid and well documented with methodological reference to extensive previous work by some of the authors.

The sample preparation and precise ^{14}C measurement by AMS of lipid samples is straightforward and well documented in the paper. In this respect the paper is up to standard for publication with minor revisions suggested in the following regarding interpretation and presentation of the results.

1. Since the reported assessments of the body fat turnover is based on measured (modelled) apparent tissue ages of the order of only 2-3 years by comparison with the atmospheric D^{14}C curve, any lag in the dietary uptake of atmospheric ^{14}C through the food chain and food storage will cause a substantial relative offset in the measured apparent tissue age, which will appear too great. Similarly, a component of marine diet will lower the tissue ^{14}C level, which will have the opposite effect. The influence of these factors is reduced because the focus of the report is on differences between different tissues and the statistical material is impressively extensive, but the issue needs to be dealt with either in the main text or supplement, especially as even the most significant resulting tissue type age differences are only of the order of 0.5 years.

2. I find Fig. 1B very confusing. The Y-axis is designated D^{14}C , but it only makes sense if the data points represent the difference $\text{D}^{14}\text{C}(\text{sample}) - \text{D}^{14}\text{C}(\text{curve})$. And yet, the high difference values in Fig. 1B up to 80 ‰ are not seen in Fig. 1C. Please clarify in the caption and on the Y-axis what is actually shown in the figure – clearly it cannot be "Individual ^{14}C lipid values related to the subjects year of birth". The same issue applies to main text lines 87 – 89. Also it is misleading to

state that the lipid age is given by this difference. Rather, it is the time difference between the biopsy collection date and the date at which the atmospheric curve was at the same ^{14}C level as that of the biopsy. This should be illustrated and explained in Fig./caption 1C.

3. Terminology: Both figure plots and the text line 74 in Online Methods state correctly that " ^{14}C data are $^{14}\text{C}/^{12}\text{C}$ ratios reported as decay-corrected $\Delta^{14}\text{C}$ in per mille deviation from a standard". " $\Delta^{14}\text{C}$ " should be used throughout the main text rather than the misleading use of "delta ^{14}C ".

The non-capitalised $\delta^{14}\text{C}$ has a different definition from $\Delta^{14}\text{C}$.

4. Acknowledgements p. 9 line 186: Should be "accelerator mass spectrometry" rather than "accelerated..."

With kind regards,

Jan Heinemeier

Aarhus 11 September 2016

REBUTTAL TO MS NCOMMS-16-19534-T**Reviewer 2**

We are very grateful for the constructive comments which have significantly improved the MS.

- *QUESTION 1) In the abstract, a link between visceral triglyceride storage capacity and/or ineffective SC storage with metabolic consequences should be included into the "results" part.*

Answer: This has been considered in the revised Abstract. However, as the abstract needed to be markedly shortened we have used a somewhat different wording than suggested.

- *QUESTION 2) The authors show that all 14C values were positioned above the atmospheric curve suggesting that the lipids were older than the collection date. Is there another way to support that assumption, e.g. by testing circulating blood cell 14C incorporation in parallel to the adipose tissue measurements? Maybe it would be sufficient to demonstrate differences between "short lived" and "older" cells in a small subgroup.*

Answer: We thank the reviewer for this suggestion. In fact, in a parallel study we had retained paired blood and fat samples from the same patients (n=22) which were collected during a time span similar to that of the present study (2010-2012). These samples have now been analyzed and demonstrate that lipid age is significantly older than blood samples from the same subject. This new data is included in the revised manuscript and forms part of the new Figure 1 (Fig. 1B, 1C). Additional information is given in a new Supplementary Table 1.

- *QUESTION 3) The statement that there were no or only weak positive linear correlations between BMI or WHR and lipid age is difficult to follow, give that no data are shown and the p-values indicate at least in part formal significance. The authors should clarify that.*

Answer: We are sorry for the vague message about correlations. . In the revised MS (new Table 1) we present (for the whole cohort) the r-values for a simple (i.e. linear) correlation between lipid age in

either adipose region and either BMI, waist-to-hip ratio or subject age. These results are thoroughly described in the Results of the revised MS.

- *QUESTION 4) It would be interesting to see, whether categorization of individuals by lipid turnover rate unravels (e.g. upper and lower 10%) associations with distinct phenotypes, such as obesity-related metabolic alterations, fat distribution, BMI etc.*

Answer: We thank the reviewer for this important comment which was not dealt with in the original MS. We have now divided the material according to the distribution of lipid age. Unfortunately, using the 10th percentile yielded too few study subjects so we have used the upper and lower 25th percentile. We would like to point out, as stated in the original MS, that due to the difficulties in recruiting study subjects we have clinical details relevant for the question only in a limited number of subjects and just for anthropometric and DEXA measurements. The findings are shown in new Table 2, mentioned in the Results and commented on in the Discussion of the revised MS. They show that whereas upper and lower quartiles of subcutaneous lipid age have similar phenotypes this is not the case for the quartiles of visceral (i.e. omental) lipid age. Subjects with low omental lipid age (lower 25th percentile) have a significantly more pernicious anthropometric phenotype than those with a high omental lipid age (upper 25th percentile).

- *QUESTION 5) Metabolically healthy obesity has been indirectly linked to SC AT expandability. Is there any way to use the data to test this hypothesis in the context of metabolically healthy versus unhealthy obese?*

Answer: This is also an important question that was not addressed properly in the original MS. There is no unanimous definition of SC WAT expandability in the literature. Nevertheless, most experts refer to impaired expandability as the inability of fat cell size and number to increase properly during development of overweight/obesity. Fortunately, we have data on subcutaneous fat cell size and sufficient clinical information to be able to classify a relatively large number of obese subjects (n=181) in terms of relative health according to ATPIII criteria and quartiles of fat cell volume. The results are presented in new Figure 5 as well as in Supplementary Table 2 and commented on in the

revised Discussion. Subjects were classified as healthy (ATPIII score of 0-2) or unhealthy (ATPIII score 3-5). There was no significant interaction between lipid age and quartiles of fat cell size in either group. As expected, unhealthy obese displayed adipose hypertrophy (few but large fat cells) in comparison to healthy subjects with similar BMI and abdominal subcutaneous fat mass. Interestingly, however, the unhealthy obese with smaller fat cells (two lowest quartiles) had significantly higher lipid age (i.e. decreased turnover) than healthy obese who were grouped into the lower two quartiles of fat cell size. This could not be explained by differences in fat cell size between healthy and unhealthy obese. In our opinion these novel data bring an additional dimension to SC AT expandability. It is not only fat cell size and number that matters but also lipid turnover in the lower range of fat cell sizes.

Additional comment: Because of journal style formatting, we have had to add a separate Discussion section. In order to make it informative we have added some comments/discussion points that were not included in the previous version of the MS. Furthermore, the Methods section has now been included in the main paper

REVIEWER 3

We thank the reviewer for the thoughtful comments which have been a great help in improving the clarity and quality of our paper.

- *QUESTION 1. Since the reported assessments of the body fat turnover is based on measured (modelled) apparent tissue ages of the order of only 2-3 years by comparison with the atmospheric $\Delta 14C$ curve, any lag in the dietary uptake of atmospheric $14C$ through the food chain and food storage will cause a substantial relative offset in the measured apparent tissue age, which will appear too great. Similarly, a component of marine diet will lower the tissue $14C$ level, which will have the opposite effect. The influence of these factors is reduced because the focus of the report is on differences between different tissues and the statistical material is impressively extensive, but the issue needs to be dealt with either in the main text or supplement, especially as even the most significant resulting tissue type age differences are only of the order of 0.5 years.*

Answer: The reviewer raises a very valid point about the possible influence of dietary lag (i.e. time delay between production and consumption of food) and dietary composition (marine versus plant and/or meat based diets) on $\Delta^{14}\text{C}$ values of biological samples. As the reviewer points out, diet does affect the isotopic ratio. However, for this to be significant the subjects must have followed unusual and extreme diets, dominated for example, by fresh fish. Marine diets are extremely rare in the presently studied urban population living in Stockholm, Sweden. Admittedly, this may occur among people living in the most distal parts of Stockholm archipelago, however such persons were not included in this study. Moreover, whilst there is no doubt that there is a variability in diet between subjects, which may contribute to some of the dispersion of values for lipid age, it should be overcome by the very large number of subjects included in the study ($n=381$). We could also add that subjects coming to our laboratory for clinical studies are normally asked about their eating habits according to a so called BITE questionnaire. Among >1400 previously investigated subjects, none have reported a strict or almost exclusive marine diet. These observations are supported by a recent study of a national dietary survey in Swedish adults (BJN, 2016). No important fraction of subjects eating a diet of predominantly marine products was found in this national population based study ($n=1740$).

Also of note, the effect of a marine diet is more noticeable when the difference between atmospheric and marine $\Delta^{14}\text{C}$ are large. Post year-2000 the effect of the marine isotopic ratio becomes less and less significant (Figure 1a, Figure 2: Georgiadou and Stenström, Radiocarbon, 2010). All samples for this study were collected between 1996-2013, where the difference between atmospheric and marine $\Delta^{14}\text{C}$ values are small. Any study using samples taken near the bomb-peak (ca. 1960-1970) would however, suffer from the aforementioned dietary effect. To get this into perspective, the difference between the atmospheric and marine $\Delta^{14}\text{C}$ values are directly related to the effect of diet on our experimental values. In 1965, this value is in the order of 800 permille and this is estimated to have a dietary effect of up to 2.5 years for a marine-dominated diet (Georgiadou & Stenström 2010). In the year 1996-2013 (the period we collected the samples), this difference has decreased by a factor varying between 18

and 40 (ca 50-20 per mille). Consequently, the estimated effect of an extreme marine diet (which we do not have) would maximally be in the range of 3-7 weeks, depending on the date of collection.

Finally, as pointed out by the reviewer, whilst the questions raised are valid and important from a methodological point of view, they are not major issues for the interpretation of the present results as we are comparing tissues from the same subjects, not in-between subjects. This applies similarly to the question of a time lag in the dietary uptake of atmospheric ^{14}C through the food chain: This lag would be expected to be the same for an individual when comparing visceral vs subcutaneous fat tissues. All these considerations are dealt with in the Methods, Results and Discussion of the revised MS and the relevant references have been added.

- QUESTION 2. I find Fig. 1B very confusing. The Y-axis is designated $\Delta 14\text{C}$, but it only makes sense if the data points represent the difference $\Delta 14\text{C}$ (sample)- $\Delta 14\text{C}$ (curve). And yet, the high difference values in Fig. 1B up to 80 ‰ are not seen in Fig. 1C. Please clarify in the caption and on the Y-axis what is actually shown in the figure – clearly it cannot be “Individual ^{14}C lipid values related to the subject’s year of birth”. The same issue applies to main text lines 87 – 89. Also it is misleading to state that the lipid age is given by this difference. Rather, it is the time difference between the biopsy collection date and the date at which the atmospheric curve was at the same ^{14}C level as that of the biopsy. This should be illustrated and explained in Fig./caption 1C.*

Answer: In retrospect we completely agree that comparing Figures 1B and 1C (in the original MS) can be confusing, as we used 1) different abscissa (year of birth vs year of collection), 2) different ranges on the abscissa and 3) different sets of data from the three different collections times (years 1996, 2003 and 2013). We apologize for being ambiguous and gratefully made the needed changes for clarity as described below.

Figure 1A remains unchanged, where we make the point that there are no long-lived pool of lipids

and that lipid age is independent of the subjects' date of birth. The data is shown for lipid samples collected after 2010. We have described the method briefly in the figure legend to and in somewhat more detail in the Results. Finally we have added a new supplement where details of the modelling are given

Fig 1B has been replaced with a new figure to illustrate the following: 1) the measured $\Delta^{14}\text{C}$ values depend on the sample collection date and 2) The values are over the bomb-peak curve, i.e., older than the collection date. We also present the measured $\Delta^{14}\text{C}$ value for paired samples of blood and subcutaneous adipose tissue from the same subjects. In new Fig 1C we confirm that the average age of blood (1 year) is significantly lower than that of fat (2.5 years) for a variety of collection dates. See also text in the revised MS.

- *QUESTION 3 Terminology: Both figure plots and the text line 74 in Online Methods state correctly that “ ^{14}C data are $^{14}\text{C}/^{12}\text{C}$ ratios reported as decay-corrected $\Delta^{14}\text{C}$ in per mille deviation from a standard”. “ $\Delta^{14}\text{C}$ ” should be used throughout the main text rather than the misleading use of “delta ^{14}C ”. The non-capitalised $\delta^{14}\text{C}$ has a different definition from $\Delta^{14}\text{C}$.*

Answer: We thank the reviewer for pointing this out. This is corrected in the revised manuscript. Note to the editor: as we passed the manuscript from author to author, different computers changed the $\Delta^{14}\text{C}$ to D^{14}C , so this is something that has been thoroughly checked in revising the work.

QUESTION 4. Acknowledgements p. 9 line 186: Should be “accelerator mass spectrometry” rather than “accelerated...”

Answer: This is corrected in the revised MS.

Reviewer #2 (Remarks to the Author)

All my comments have been perfectly addressed by the authors. I do not have any further points.

Reviewer #3 (Remarks to the Author)

I am pleased to report that I find that the authors have very satisfactorily dealt with all of my comments, suggestions and points of criticism in their rebuttal and their revised manuscript. From my point of view the manuscript is now acceptable for publication.

With kind regards,

Jan Heinemeier

Aarhus 16 January 2017